# Fungi-based food in the public eye: Terminology, cultivation timelines, sustainability, and nutrition across three EU countries

Coralie Hellwig[1,2], Mohammad J. Taherzadeh[1,2]*

1 Swedish Centre for Resource Recovery, University of Borås, Borås, Sweden, 2 Millow AB, Gothenburg, Sweden

* mohammad.taherzadeh@hb.se

## Abstract

Fungi-based foods are gaining attention as sustainable alternatives, yet public understanding remains limited. This study quantified knowledge and perceptions in Germany, Spain, and Sweden, selected for their distinct dietary cultures and regulatory contexts. An online survey of 6,004 adults used quiz-style questions covering taxonomy, terminology, cultivation timelines, sustainability, and nutrition. Results show misconceptions. Many classified fungi as plants, were unfamiliar with mycoprotein, and underestimated the rapid cultivation of filamentous fungi. Despite these gaps, majorities in all countries endorsed fungi's sustainability potential and nutritional parity with meat. Greater knowledge was observed among tertiary-educated, higher-income, and meat-reducing groups. These findings show that limited process-related factual knowledge exists, underscoring the need for clearer communication. Public messaging should employ accessible terminology, present fermentation as 'grown in days,' and emphasize nutritional properties. Addressing knowledge gaps can align perceptions with evidence, enable informed decision-making, and support integration of fungi-based foods into sustainable systems.

## Introduction

Fungi-based food, i.e., foods produced from filamentous fungal mycelium via aerobic biomass fermentation, has re-emerged as a promising component of sustainable, nutritious diets, with several recent reviews documenting its production biology, nutritional attributes, and market evolution [1–3]. In parallel, modeling and life cycle assessment studies indicate that fungi-based food can contribute to more sustainable food systems, e.g., by sparing land and reducing deforestation-related emissions, and by lowering climate impacts compared to animal-derived meat [4,5]. Beyond environmental performance, human nutrition studies increasingly support the positive nutritional offering and the parity with high-quality animal proteins [6].

**Data availability statement:** All relevant data are within the paper and its Supporting Information files.

**Funding:** Swedish Research Council FORMAS with grant number 2023-02018.

**Competing interests:** The authors have declared that no competing interests exist.

Conceptually, this study assesses domain-specific factual knowledge and semantic familiarity related to fungi-based foods, rather than literacy as a generalized competency. The five survey statements are treated as single-item indicators of distinct facets: (i) taxonomy classification; (ii) semantic familiarity with the term 'mycoprotein', noting variable regulatory and market usage; (iii) process-related factual knowledge about cultivation timelines; and (iv–v) belief-like agreement with positively framed sustainability and nutrition claims. Accordingly, the statements are not interpreted as a psychometrically validated literacy scale or as evidence of deeper conceptual mastery or functional application.

Despite encouraging progress, many remain unfamiliar with the characteristics and benefits of fungi-based foods [7]. For instance, the term *mycoprotein* may be misunderstood or entirely unknown, and the speed and efficiency of fungal cultivation may not be widely recognized. This knowledge gap can lead to misconceptions and hesitancy, thereby hindering the potential of fungi-based alternatives to be fully realized in addressing global sustainability goals [8]. Cross-disciplinary reviews on new food technologies show that low familiarity and terminology can shape evaluation and acceptance well before taste or price are considered [9].

Building on prior work on perceptions and engagement with fungi-based foods, e.g., [8,10], the current study provides a large-scale, cross-national quantification of specific factual knowledge gaps related to terminology, cultivation timelines, sustainability claims, and nutrition. This was done through an online survey in Germany, Spain, and Sweden, examining recognition of basic factual concepts, e.g., distinctions between fungi, mushrooms, and plants, and process and product attributes relevant for public communication. In this way, the present study complements existing qualitative and attitudinal research by providing baseline prevalence estimates of specific misconceptions and knowledge gaps that may shape how fungi-based foods are evaluated in public discourse. The three countries were chosen for their distinct dietary cultures, varying perceptions of food sustainability, and diverse regulatory frameworks [11–13]. How knowledge varies across sociodemographic and dietary segments identified in prior work as important for alternative-food engagement is also explored.

By identifying gaps in public understanding, this study was aimed to inform the development of transparent communication strategies that support the provision of education and thereby enables informed decision-making. Ultimately, these insights can help guide stakeholders in harnessing fungi-based foods as a sustainable, scalable food source, contributing to food security initiatives and aligning with policy directives such as the EU's Farm to Fork Strategy.

## Materials and methods

### 2.1. Data collection

The study assessed respondents' domain-specific factual knowledge and semantic familiarity related to fungi-based foods using five closed-ended statements. These statements cover taxonomy, familiarity with the term mycoprotein, cultivation timelines, and agreement with broad sustainability and nutrition. Because the instrument

was designed for large-scale, cross-national fielding, the statements were intentionally brief and are interpreted as indicators of knowledge gaps or claim-related beliefs, not as a comprehensive literacy measure. The English master questionnaire, including the exact item wording, response options, and real-time feedback, and the professionally translated German, Swedish and Spanish versions thereof, are provided in Appendix 1.

Interpretation of several statements warrants additional caution. The taxonomy statement 'Fungi and mushrooms are the same thing and are plants' combines two propositions, namely the classification of fungi versus plants and the relationship between fungi and mushrooms; therefore, responses reflect a composite misconception and cannot distinguish which proposition drove an incorrect answer. The mycoprotein statement relied on a simplified, survey-appropriate definition; because mycoprotein is used in regulatory and market contexts with some variability, this item should be interpreted as assessing familiarity with a common usage rather than adjudicating a single authoritative definition. The cultivation-time statement uses the term 'filamentous fungi,' which may be unfamiliar to some respondents. Some may map the question onto mushroom cultivation timelines, meaning that incorrect or uncertain responses may partly reflect semantic ambiguity rather than a precise misconception about fermentation-based cultivation cycles. Finally, the sustainability and nutrition statements are positively framed declarative statements; high agreement may capture general beliefs or openness to claims, and may be susceptible to acquiescence and ceiling effects, rather than deep, tested understanding.

## 2.2. Panel recruitment

E-mail invitations were issued to pre-profiled panel members aged ≥18 years. The opt-in panel is assembled through multiple channels such as pay-per-click advertising, affiliate networks, and referrals to maximize demographic and attitudinal diversity. Quality controls included domain/IP screening, CAPTCHA and double-opt-in verification, and post-enrolment checks including consistency tests, geolocation validation, duplicate detection, and attention checks. Participation was compensated via a modest points-based incentive redeemable for cash or vouchers to discourage low-quality responses. As with other online-only panels, nonprobability recruitment and repeated survey participation may introduce selection effects and panel conditioning; therefore, prevalence estimates should be interpreted as applying to the survey samples rather than as population parameters. Informed consent was obtained in accordance with ESOMAR/GDPR in writing. The study protocol was approved by the Swedish Ethical Review Authority (approval no. 2024-04842-01).

## 2.3. Country-level and sociodemographic variables

National context can shape exposure to media, policy environments, and cultural familiarity with fungi-based foods and prior research suggests that younger adults more often engage with emerging food trends and alternative foods [14]; women tend to report greater interest in meat reduction than men [15]; people already using meat alternatives, e.g., vegetarians, flexitarians, may possess higher baseline knowledge about fungi-based foods [16]; higher income and education are associated with greater access to scientific and environmental information [17]; urban residence increases contact with new food products [18]; and culinary traditions involving fermentation or fungi, e.g., for tempeh, can foster familiarity [19]. Therefore, country and standard sociodemographic characteristics, i.e., gender, age, education, household income, ethnic background, dietary preference, and geography, were examined. The sociodemographic profile of the respondents is presented in Appendix 2.

## 2.4. Data analyses

Responses were analyzed quantitatively. Descriptive statistics summarized the distribution of answers for each question. To evaluate the hypothesis that the proportions selecting the correct answers regarding filamentous fungi in terms of taxonomy, cultivation, nutritional attributes, and resource-efficiency differ across Germany, Spain, and Sweden, tests of differences in proportions were conducted. Column-proportion tests were computed for each relevant pair of country columns within each table row. Test statistics ($t$), associated degrees of freedom, and $p$-values were derived for each comparison;

statistical significance was set at $p < 0.05$. Additional analyses assessed whether sociodemographic factors were associated with knowledge. All statistical analyses were performed by YouGov.

To complement tests of statistical significance, effect sizes were calculated for all subgroup comparisons reported in the manuscript. For categorical comparisons based on contingency tables, Cramer's V was computed as a measure of association. For selected binary outcomes, odds ratios (ORs) with 95% confidence intervals were additionally estimated using logistic regression models, contrasting subgroup levels against reference categories. Given the large number of subgroup comparisons, $p$-values were adjusted for multiple testing using the False Discovery Rate (FDR) control according to the Benjamini–Hochberg procedure. Adjustments were applied within families of related tests, i.e., outcomes within the same subgroup and country. Subgroup analyses are therefore interpreted as exploratory, with emphasis placed on patterns that remain robust after FDR correction. Effect size estimates and adjusted $p$-values are reported in the Supporting Information (Appendix 3).

## Results and discussion

This study was designed to quantify public knowledge about fungi-based foods across Germany, Spain, and Sweden, and to identify cross-national and sociodemographic patterns in responses. Because the study is cross-sectional and does not measure mediators such as biology coursework, media exposure, or product familiarity, observed patterns are reported descriptively. Where potential mechanisms are discussed, these are presented explicitly as hypotheses for future research rather than as conclusions.

### 3.1. Perceptions of fungi: conceptions about taxonomy and classification

The level of knowledge of the distinction between fungi and mushrooms and plants was explored because misconceptions about fungi as plants is common [20]. Fungi and plants belong to distinct taxonomic kingdoms and differ fundamentally in their biology. While plants (Kingdom Plantae) are autotrophic organisms that produce energy through photosynthesis and have cell walls composed of cellulose, fungi (Kingdom Fungi) are heterotrophic, absorb nutrients from organic matter, and possess cell walls made of chitin or chitosan [21]. The molecular evidence shows that fungi are evolutionarily more closely related to animals than to plants.

In response to the statement 'fungi and mushrooms are the same thing and are plants,' 60% of German respondents answered 'true,' compared to 38% of Spanish respondents and 34% of Swedish respondents. More respondents in Spain (62%) and Sweden (66%) selected 'false' (Fig 1).

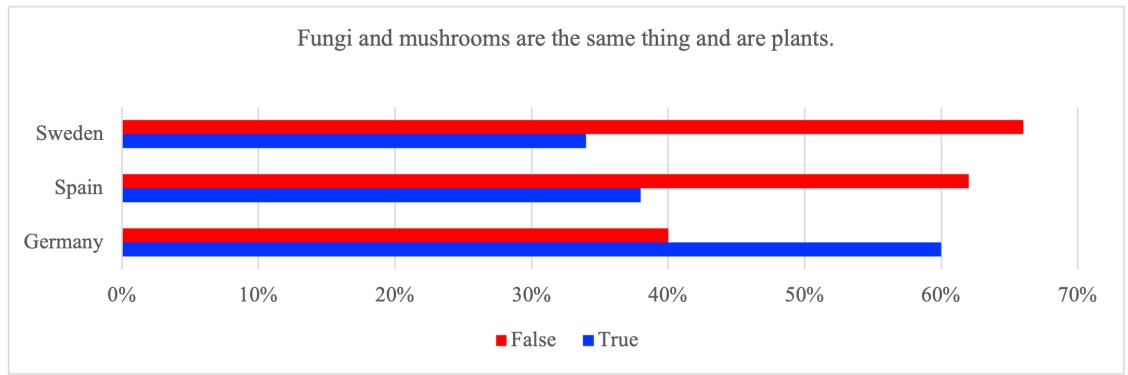

**Fig 1. Proportion of respondents in Germany, Spain and Sweden who answered 'True' or ´False' to the statement ´Fungi and mushrooms are the same thing and are plants.**

Because this statement intentionally combines two propositions, i.e., whether fungi are plants and whether mushrooms are the same thing as fungi, it captures a composite misconception. Accordingly, incorrect responses cannot be attributed to one specific misunderstanding, and results are interpreted as indicating a general taxonomy-related knowledge gap rather than a precise diagnosis of which concept is confused.

Statistical analyses show that the proportion of German respondents selecting the correct answer is significantly lower, whereas those from Spain and Sweden are significantly higher (Table 1). An omnibus association between country and response was statistically significant ($\chi^2$=318.0, p_FDR<0.001), with Cramer's V=0.23, indicating a small-to-moderate association.

The magnitude of the German result suggests that national-level factors beyond a simple cohort effect may also contribute. Given that hypothetically influential aspects such as legacy educational framings and public discourse that historically grouped fungi with plants may persist differently across countries, further research is needed to understand this finding. The present study did not include cognitive interviewing or formal measurement-invariance testing, so future work should also explicitly evaluate cross-language comparability or other potential mechanisms behind national differences. Beyond country-level variation, there are significant demographic influences. Younger adults (18–24 years old) in Germany and Spain were more likely to answer correctly that fungi are not plants, whereas older adults (55+ years) in these countries were more prone to the misconception (Table 2).

To assess the magnitude of these subgroup differences, effect sizes were calculated for all reported comparisons. The association between country and correct classification of fungi was of small-to-moderate magnitude (Cramer's V), indicating meaningful but not large differences. Education-related differences showed the most consistent effects, with respondents with tertiary education displaying higher odds of correctly identifying fungi as non-plants compared with those with primary education. After adjustment for multiple comparisons using FDR control, education-related patterns remained robust, whereas several age- and region-specific differences should be interpreted as exploratory (see Supplementary Tables in Appendix 3).

An age gradient was observed for correct identification of fungi, with older respondents more likely to endorse the taxonomy misconception (Table 2). One plausible yet hypothetical interpretation is that older cohorts may have been exposed to earlier biology curricula or popular framings that treated fungi as a subset of plants; however, because such cohort effects could plausibly operate across countries, we do not treat this as a definitive explanation for the stronger German pattern. Instead, we present cohort-related educational change as one hypothesis among several that should be tested directly in future work using measures of science coursework, media exposure, and cross-language cognitive testing.

**Table 1. Rating profiles and statistically relevant results of the true-or-false statement 'fungi and mushrooms are the same thing and are plants' across Germany (DE), Spain (ES) and Sweden (SE)*.**

|  | Total | Country | | |
|---|---|---|---|---|
|  |  | DE | ES | SE |
|  |  | A | B | C |
| Base | 6004 | 2002 | 2002 | 2000 |
| True | 44% | 60% | 38% | 34% |
|  |  | ▲ B.C | ▼ C | ▼ |
| False | 56% | 40% | 62% | 66% |
|  |  | ▼ | ▲ A | ▲ A.B |
| Total Sum | 100% | 100% | 100% | 100% |

*Cell Contents (Column Percentages, Statistical Test Results), Statistics (Column Proportions, (95%): A/B/C, Minimum Base: 30 (**), Small Base: 100 (*))

▲ indicates result is significantly higher than the result in the Total column.

▼ indicates result is significantly lower than the result in the Total column.

**Table 2. Statistically significant sociodemographic variables regarding the correct answer to the true-or-false statement 'fungi and mushrooms are the same thing and are plants'.**

| Country* | Sociodemographic variable | Subgroup | Direction of significance** |
|---|---|---|---|
| DE, ES | Age | 18–24 years | ▲ |
| DE | Region | Berlin, Brandenburg, Hessen, Thuringia | ▲ |
| ES | Education | Tertiary | ▲ |
| DE, ES | Age | 55 + years | ▼ |
| SE | Age | 18–24 years | ▼ |
| SE | Region | Scania | ▼ |
| DE, ES | Education | Primary | ▼ |
| ES | Diet | Pescatarian | ▼ |
| SE | Diet | Vegan, vegetarian | ▼ |
| DE | Ethnicity | East Asian | ▼ |
| ES | Ethnicity | 'Don't know' | ▼ |

*Germany (DE), Spain (ES), Sweden (SE).

**Statistical Test Results.

▲ indicates result is significantly higher than the result in the Total column.

▼ indicates result is significantly lower than the result in the Total column.

Educational attainment was also associated with accuracy in some countries (Table 2). This association could reflect differences in exposure to formal biological classification, science engagement, or information-seeking behaviors, but the survey did not measure these pathways. Moreover, because the taxonomy item combines two propositions, subgroup differences cannot be attributed to a single underlying misconception. We therefore interpret education-related patterns cautiously and recommend that future studies use separate taxonomy items and directly measure explanatory factors.

Associations with dietary self-identification were mixed. Notably, in Sweden, vegans and vegetarians showed lower accuracy on the taxonomy item, which contrasts with their generally higher performance on several other knowledge items in this survey. Given the large number of subgroup comparisons and the small effect sizes, this may be interpreted as an isolated, exploratory result rather than evidence of a systematic diet-related pattern. Future work should examine whether interpretations of the compound wording differ by dietary subgroup using cognitive interviewing or experimental item designs.

## 3.2. Understanding of the term mycoprotein

Mycoprotein is a food ingredient derived from filamentous fungi and should not be confused with fungi themselves [3]. While the term mycoprotein is often used to refer specifically to biomass obtained when fungi are cultivated under controlled fermentation conditions, the term actually refers to the protein of fungi. Given that the term 'mycoprotein' is used in the marketplace, the second question examined whether participants know what this is. It was speculated that the concept of mycoprotein may be poorly understood, with many unable to distinguish between the organism (fungi and its mycelium) and the ingredient (mycoprotein).

The survey item operationalized mycoprotein using the simplified, lay description 'proteins in fungi and mushrooms' to enable cross-national fielding. However, the term is also used variably across regulatory, branding, and market contexts, e.g., sometimes referring to a specific fermented fungal biomass ingredient or product category. Responses were therefore primarily treated as indicating familiarity with a commonly used label and its broad biological origin, rather than as a definitive test of knowledge of a single standardized definition.

                                              

In all three countries, a majority of respondents provided an incorrect definition of the term mycoprotein or indicated that they did not know what it is. In Germany, 33% correctly identified mycoprotein as 'proteins in fungi and mushrooms,' while 40% were unsure, leading to a combined 67% net incorrect. Spanish respondents showed a similar pattern (32% correct, 38% 'don't know,' 68% net incorrect). Sweden had the highest share of correct identifications at 37%, but 38% remained uncertain, and 63% were ultimately incorrect (Fig 2). The proportion of Swedish respondents selecting the correct answer is significantly higher (Table 3). Country differences in response patterns were statistically significant ($\chi^2$=113.9, p_FDR<0.001), but the association was small in magnitude (Cramer's V = 0.10).

Several subgroup findings are statistically relevant (Table 4). The proportions of Swedish male respondents; respondents from all three countries with tertiary education; respondents from all three countries with higher household incomes; both German and Swedish respondents who follow a flexitarian diet; and Swedish respondents who follow a vegan diet selecting the correct answer are significantly higher. Conversely, significantly lower proportions were observed among Swedish female respondents; 55 + year-old German respondents; Swedish respondents from Jönköping County; respondents from all three countries with primary education; respondents from Germany selecting 'other' for their diet; Spanish respondents selecting 'other' and 'don't know' for their diet; as well as German respondents who identify their ethnic background as East Asian and respondents selecting 'don't know' for their ethnicity.

Effect size estimates further clarify these subgroup differences. Associations between education, income, and correct identification of mycoprotein were consistently of small-to-moderate magnitude (Cramer's V), while odds ratios from logistic regression indicated substantially higher likelihood of correct identification among respondents with tertiary education and higher household incomes. These effects remained statistically robust following FDR correction. In contrast, several diet- and region-specific subgroup differences attenuated after FDR adjustment and are therefore interpreted as exploratory (see Supplementary Tables in Appendix 3).

These patterns suggest that overall awareness of mycoprotein's fungal origin remains limited, despite the term's presence in the marketplace. One possible reason for the misunderstanding is current labeling practices, which can imply an isolated protein fraction rather than the broader fungal mycelium typically used. A recent application for determination of

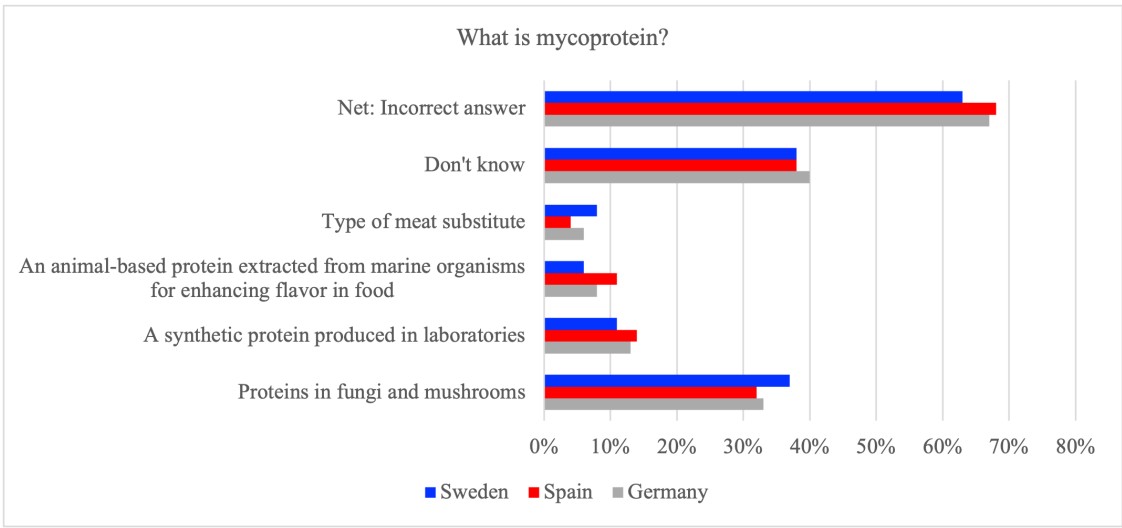

**Fig 2. Proportion of respondents in Germany, Spain and Sweden selecting what they think mycoprotein is, including the aggregated percentage of incorrect answers ('Net: Incorrect answer').**

**Table 3. Rating profiles and statistically relevant results to the question 'what is mycoprotein?' across Germany (DE), Spain (ES) and Sweden (SE)\*.**

| | Total | Country | | |
| --- | --- | --- | --- | --- |
| | | DE | ES | SE |
| | | A | B | C |
| Base | 6004 | 2002 | 2002 | 2000 |
| Proteins in fungi and mushrooms | 34% | 33% | 32% | 37% |
| | | | | ▲ A.B |
| A synthetic protein produced in laboratories | 13% | 13% | 14% | 11% |
| | | | C | |
| An animal-based protein extracted from marine organisms for enhancing flavor in food | 8% | 8% | 11% | 6% |
| | | C | ▲ A.C | ▼ |
| Type of meat substitute | 6% | 6% | 4% | 8% |
| | | B | ▼ | ▲ A.B |
| Don't know | 39% | 40% | 38% | 38% |
| | | | | |
| Net: Incorrect answer | 66% | 67% | 68% | 63% |
| | | C | C | ▼ |
| Total Sum | 100% | 100% | 100% | 100% |

\*Cell Contents (Column Percentages, Statistical Test Results), Statistics (Column Proportions, (95%): A/B/C, Minimum Base: 30 (\*\*), Small Base: 100 (\*))

▲ indicates result is significantly higher than the result in the Total column.

▼ indicates result is significantly lower than the result in the Total column.

**Table 4. Statistically significant sociodemographic variables regarding the correct answer to the question 'what is mycoprotein?'.**

| Country\* | Sociodemographic variable | Subgroup | Direction of significance\*\* |
| --- | --- | --- | --- |
| SE | Gender | Male | ▲ |
| DE, ES, SE | Education | Tertiary | ▲ |
| DE, ES, SE | Household income | Higher | ▲ |
| DE, SE | Diet | Flexitarian | ▲ |
| SE | Diet | Vegan | ▲ |
| SE | Gender | Female | ▼ |
| DE | Age | 55+years | ▼ |
| SE | Region | Jönköping | ▼ |
| DE, ES, SE | Education | Primary | ▼ |
| DE, ES | Diet | 'Other' | ▼ |
| ES | Diet | 'Don't know' | ▼ |
| DE | Ethnicity | East Asian | ▼ |
| DE, ES, SE | Ethnicity | 'Don't know' | ▼ |

\*Germany (DE), Spain (ES), Sweden (SE)

\*\*Statistical Test Results

▲ indicates result is significantly higher than the result in the Total column.

▼ indicates result is significantly lower than the result in the Total column.

novel food status under Article 4 of Regulation (EU) 2015/2283 of the European Parliament and of the Council describes mycoprotein as mycelial biomass obtained by fermentation [22].

Current discussions stress the importance of transparency and standardization in labeling for alternative foods [23]. Recent analyses and guidance discuss naming conventions and label content for mycelium products [24]. Clearer and less technical terminology, e.g., 'fungi-based' or 'mycelium-based', and supplemental educational messaging, may help reduce confusion; however, the present survey did not test labeling interventions, so any implications for willingness to engage with this kind of food should be treated as hypotheses. More generally, prior studies suggest that discomfort with mold associations can correlate with unmeasured attitudinal factors, e.g., food technology neophobia, whereas familiarity with edible fungi or fermented foods is associated with higher acceptance [25]. Future experimental work could test whether specific framings, e.g., analogies to brewing or leavening, improve understanding and perceived naturalness and whether these mechanisms mediate willingness to try fungi-based foods.

Sweden showed a higher proportion of respondents selecting the taxonomy response coded as correct. This cross-national difference may reflect a combination of measurement and contextual factors, such as differences in translation or everyday terminology for fungi and mushrooms, differential exposure to fungi-related products and media coverage, or other unobserved national-level influences. Because the survey did not include cognitive interviewing or measurement-invariance testing, these explanations cannot be evaluated here and should be treated as hypotheses for future research.

Gradients by education and income align with systematic reviews on novel foods in the EU. Higher food-technology-related knowledge and familiarity and nutrition knowledge are repeatedly associated with better comprehension, lower perceived risk, and greater openness to unfamiliar foods [26]. These subgroup effects may disentangle knowledge deficits from affective drivers of engagement and suggest that targeted education could be impactful.

Diet effects, i.e., higher correct answers among flexitarians and (in Sweden) vegans, fit multi country evidence on fungi-based food. Structural models show that willingness to try, buy, and pay a premium for this food is driven by perceived healthfulness, safety, and sustainability, with stronger effects among those who already moderate meat intake [27]. The current results extend this by indicating that not only willingness but also basic conceptual knowledge tracks with diet, possibly due to more exposure to fungi-based products and their narratives.

### 3.3. Awareness of the cultivation duration of filamentous fungi

Mushrooms are the fruiting bodies of certain fungi, typically cultivated over several weeks before they can be harvested as food [28]. In contrast, filamentous fungi have thread-like mycelium, which can be cultivated through controlled fermentation and harvested within only a few days [21]. This makes filamentous fungi highly efficient for food production, in sharp contrast to the longer and more resource-intensive cultivation of mushrooms. Filamentous fungi and mushrooms are both fungi, but they represent different products and cultivation processes that may be confused. Because respondents may assume that filamentous fungi require similar cultivation times to mushrooms, and because the term 'filamentous fungi' may itself be unfamiliar, the item likely captures both process-related factual knowledge and semantic familiarity with fermentation-based production.

Across all three countries, only a minority of respondents selected a week or less as the cultivation period for filamentous fungi, which is considered the correct timeframe. In Germany, 21% chose the correct option, 20% did so in Spain, and 25% in Sweden. The net incorrect rates were 79% in Germany, 80% in Spain, and 75% in Sweden. A substantial proportion in each country indicated uncertainty, with 'I do not know' ranging from 35% in Spain to 47% in Sweden (Figure 3).

The results suggest that overall awareness of the short cultivation time for filamentous fungi remains low across the three national survey samples, a pattern that is consistent with prior evidence that process-related factual knowledge for fungi-based food is low and that core production facts are often unfamiliar [7,25,27]. While the pattern of incorrect responses is broadly consistent across the three countries, Sweden shows a slightly higher percentage of correct answers but also the highest share of respondents who expressed uncertainty. The proportion of Swedish respondents selecting

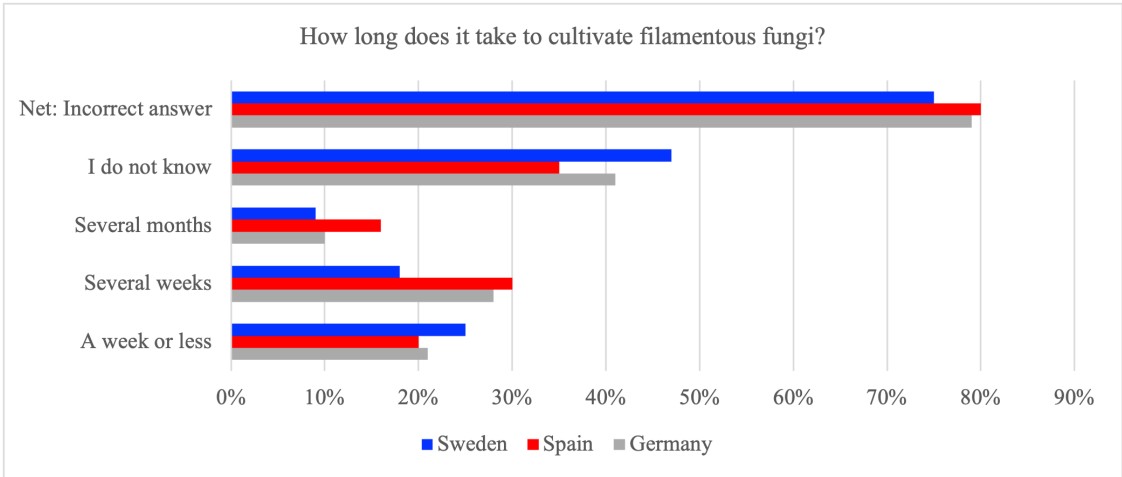

**Fig 3. Proportion of respondents in Germany, Spain, and Sweden who selected the time period for cultivating filamentous fungi which they thought was correct, including the aggregated percentage of incorrect answers ('Net: Incorrect answer').**

the correct answer is significantly higher, while that of Spanish respondents is significantly lower (Table 5). An omnibus country-by-response association was statistically significant ($\chi^2$ =209.0, p_FDR < 0.001), with a small effect size (Cramer's V = 0.13).

Subgroup analyses (Table 6) revealed that significantly higher proportions of Swedish male respondents; 25–34-year-old and 35–44-year-old German respondents; both German and Swedish respondents with tertiary education; German respondents with middle household incomes; German and Swedish respondents with higher household incomes; German respondents who follow a vegan diet; and Spanish respondents who follow a vegetarian diet selected the answer 'a week or less'. Significantly lower proportions of Swedish female respondents; 55 + year-old German respondents; Swedish respondents from Gavleborg County; both Swedish and German respondents with primary education; German respondents with lower household incomes; German respondents who eat meat; Spanish respondents who selected 'don't know' for their diet; and Swedish respondents who follow a pescatarian diet selected the correct answer.

Effect size analyses indicate that differences in awareness of filamentous fungi cultivation time were modest in magnitude overall (Cramer's V), but showed consistent and meaningful gradients by education and dietary identity. Respondents with tertiary education and those following meat-reducing diets had higher odds of selecting the correct short cultivation timeframe, and these patterns remained robust after FDR adjustment. Several age-, gender-, and region-specific subgroup effects did not remain statistically robust after correction and are therefore treated as exploratory (see Supplementary Tables in Appendix 3).

The subgroup patterns align with well-studied determinants of understanding and acceptance of new food technologies. Higher accuracy among respondents with tertiary education and higher incomes aligns with evidence that food-technology knowledge is positively associated with comprehension and favorable evaluations of new foods, while lower education is linked to greater uncertainty [29,30]. Age gradients, i.e., lower accuracy among older adults, are consistent with findings that unmeasured attitudinal factors, such as food technology attitudes, tend to be higher in later life, which can dampen openness to unfamiliar production processes [31]. Differences by diet identity, i.e., higher accuracy among vegans and vegetarians and, in some countries, flexitarians, cohere with multi-country modeling showing that individuals already oriented toward meat reduction report greater willingness to try and pay for fungi-based food, possibly reflecting greater exposure to, and therefore knowledge of, fermentation-based foods [27].

**Table 5. Rating profiles and statistically relevant results to the question 'how long does it take to cultivate filamentous fungi?' across Germany (DE), Spain (ES) and Sweden (SE)\*.**

|  | Total | Country | | |
|---|---|---|---|---|
|  |  | DE | ES | SE |
|  |  | A | B | C |
| Base | 6004 | 2002 | 2002 | 2000 |
| A week or less | 22% | 21% | 20% | 25% |
|  |  |  | ▼ | ▲ A,B |
| Several weeks | 25% | 28% | 30% | 18% |
|  |  | ▲ C | ▲ C | ▼ |
| Several months | 12% | 10% | 16% | 9% |
|  |  |  | ▲ A,C | ▼ |
| I do not know | 41% | 41% | 35% | 47% |
|  |  | B | ▼ | ▲ A,B |
| Net: Incorrect answer | 78% | 79% | 80% | 75% |
|  |  | C | ▲ C | ▼ |
| Total Sum | 100% | 100% | 100% | 100% |

\*Cell Contents (Column Percentages, Statistical Test Results), Statistics (Column Proportions, (95%): A/B/C, Minimum Base: 30 (\*\*), Small Base: 100 (\*))

▲ indicates result is significantly higher than the result in the Total column.

▼ indicates result is significantly lower than the result in the Total column.

**Table 6. Statistically significant sociodemographic variables regarding the correct answer to the question 'how long does it take to cultivate filamentous fungi?'.**

| Country\* | Sociodemographic variable | Subgroup | Direction of significance\*\* |
|---|---|---|---|
| SE | Gender | Male | ▲ |
| DE | Age | 25–34, 35–44 years | ▲ |
| DE, SE | Education | Tertiary | ▲ |
| DE | Household income | Middle | ▲ |
| DE, SE | Household income | Higher | ▲ |
| DE | Diet | Vegan | ▲ |
| ES | Diet | Vegetarian | ▲ |
| SE | Gender | Female | ▼ |
| DE | Age | 55+ years | ▼ |
| SE | Region | Gavleborg | ▼ |
| DE, SE | Education | Primary | ▼ |
| DE | Household income | Lower | ▼ |
| DE | Diet | Eat meat | ▼ |
| ES | Diet | 'Don't know' | ▼ |
| SE | Diet | Pescatarian | ▼ |

\*Germany (DE), Spain (ES), Sweden (SE)

\*\*Statistical Test Results

▲ indicates result is significantly higher than the result in the Total column.

▼ indicates result is significantly lower than the result in the Total column.

These findings highlight a critical missed communication opportunity and indicate a general knowledge gap about the time scale. Edible filamentous fungi are grown in controlled, aerobic bioreactors with rapid growth kinetics and high volumetric productivity and the short growth cycle of filamentous fungi is a significant advantage over traditional meat production and many plant-based alternatives [3,32]. Communication strategies tested in adjacent literatures found that plain-language, benefit-oriented explanations improve acceptance of unfamiliar food technologies compared with highly technical framings, suggesting that explicitly stating, e.g., that fungi-based food is grown in days via controlled food-grade fermentation, can address misperceptions about time scales while building perceived naturalness and trust [9,33].

Given the observed socio-demographic gradients, segmenting messages, i.e., more explanatory depth for lower-education or higher-food technology attitudes groups, and sustainability framing for meat-reducers, aligns with evidence on heterogeneity in technology perceptions and the role of perceived benefits and risks in shaping acceptance [30,34]. Linking the brief growth period to sustainability, by stressing that shorter growth times entail less resource use, fewer inputs, and consistent product quality, could directly address environmental impact concerns if effectively communicated and become a motivation to engage with this kind of food [7,8].

### 3.4. Perceptions on filamentous fungi and sustainable resource management

Fungi-based foods are frequently communicated as environmentally advantageous compared with ruminant meat. To contextualize this claim, life-cycle and systems analyses commonly report substantially lower land use and reduced deforestation pressure when microbial proteins replace ruminant meat, although estimates vary by production assumptions and system boundaries [4,5,35]. Respondents' agreement with the statement that cultivating filamentous fungi can play a big role in managing the planet's resources was therefore explored. The results of the current study indicates that agreement with the sustainability statement was high in all three countries (Germany 85%, Spain 88%, Sweden 86%; Fig 4). Given the positively framed wording and the very high endorsement levels, this item is best interpreted as capturing general beliefs or openness to a sustainability claim rather than tested, in-depth knowledge. The high baseline agreement suggests a ceiling effect that limits discrimination across subgroups. These features also increase susceptibility to acquiescence bias, i.e., a tendency to agree with positively framed statements.

Spanish respondents endorsed the statement slightly more often than German respondents (Table 7). Given the likely ceiling and acquiescence effects and the very small effect size, attributing this difference to specific national drivers should be avoided. Rather, future research could test whether differences such as media exposure, policy salience, or

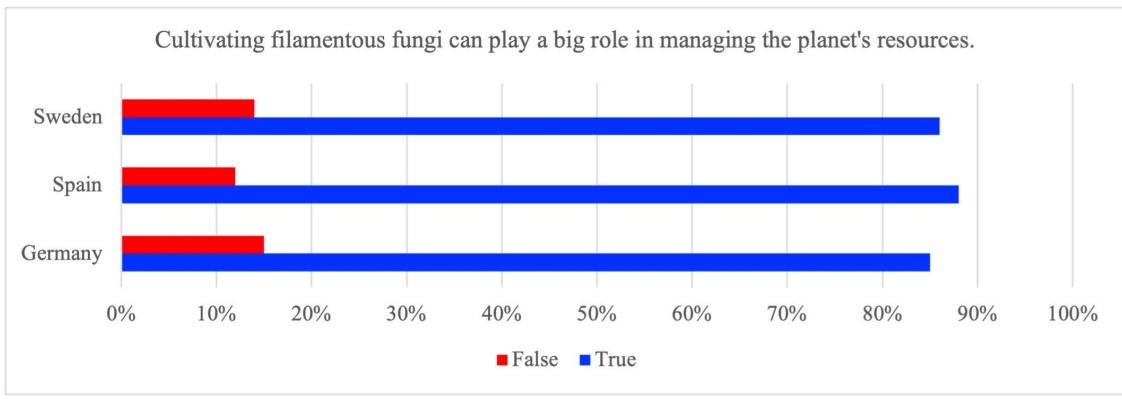

**Fig 4. Proportion of respondents in Germany, Spain, and Sweden who answered 'True' or 'False' to the statement 'Cultivating filamentous fungi can play a big role in managing the planet's resources.**

**Table 7. Rating profiles and statistically relevant results of the true-or-false statement 'cultivating filamentous fungi can play a big role in managing the planet's resources' across Germany (DE), Spain (ES) and Sweden (SE)*.**

| | Total | Country | | |
|---|---|---|---|---|
| | | DE | ES | SE |
| | | A | B | C |
| Base | 6004 | 2002 | 2002 | 2000 |
| True | 86% | 85% | 88% | 86% |
| | | | ▲ A.C | |
| False | 14% | 15% | 12% | 14% |
| | | B | ▼ | B |
| Total Sum | 100% | 100% | 100% | 100% |

* Cell Contents (Column Percentages, Statistical Test Results), Statistics (Column Proportions, (95%): A/B/C, Minimum Base: 30 (**), Small Base: 100 (*))

▲ indicates result is significantly higher than the result in the Total column.

▼ indicates result is significantly lower than the result in the Total column.

environmental discourse explain cross-national variation. Consistent with the ceiling pattern, cross-national differences were statistically detectable but very small in magnitude ($\chi^2$=7.9, p_FDR = 0.019; Cramer's V = 0.04).

Subgroup analyses (Table 8) further revealed that significantly higher proportions of German and Spanish respondents with middle household incomes; Spanish respondents with higher household incomes; German respondents who follow vegan and flexitarian diets; and Swedish respondents who follow a pescatarian diet endorsed the statement. Significantly lower proportions are observed among 18–24-year-old Spanish respondents; both German and Spanish respondents with primary education; respondents from all three countries who prefer not to disclose their household income; both German and Spanish respondents who selected 'other' for their diet; Spanish respondents who selected 'don't know' for their diet; German respondents who describe their ethnic origin as South Asians; and respondents from all three countries who selected 'don't know' for their ethnicity.

These findings are consistent with prior work reporting generally favorable sustainability perceptions toward fungi-based foods [8,10]. In the present study, high agreement should be interpreted as perceived plausibility of a sustainability claim rather than as verified understanding of environmental performance or as evidence of engagement with such food.

Education emerged as a consistent gradient in endorsement of sustainability and nutrition statements and in correct responses on taxonomy and cultivation items. This pattern aligns with broader evidence that educational attainment can act as a structural determinant of sustainability engagement. For example, a representative Slovenian study found that higher education predicted more sustainable dietary attitudes and lower meat consumption, even after controlling for confounders [36]. Related work also suggests that education may shape dietary behavior partly via competencies to access, understand, and evaluate health-related information, which can mediate socioeconomic differences in dietary outcomes [36]. These studies support interpreting the repeated education gradients observed here as reflecting differences in information access and evaluative competencies, while noting that the present survey did not measure these mechanisms directly.

Recent work also emphasizes that 'sustainability' is a property of the production system rather than the organism alone. Upstream choices, e.g., substrate sourcing, use of side-streams, and electricity mix, and downstream processing can materially shift climate and resource footprints [2, 37,38]. Accordingly, the present survey should not be read as validating specific environmental magnitudes as it merely indicates that many respondents view the general sustainability claim as credible.

**Table 8. Statistically significant sociodemographic variables regarding endorsement of the true-or-false statement 'cultivating filamentous fungi can play a big role in managing the planet's resources'\*\*.**

| Country* | Sociodemographic variable | Subgroup | Direction of significance** |
|---|---|---|---|
| DE, ES | Household income | Middle | ▲ |
| ES | Household income | Higher | ▲ |
| DE | Diet | Vegan, flexitarian | ▲ |
| SE | Diet | Pescatarian | ▲ |
| ES | Age | 18–24 years | ▼ |
| DE, ES | Education | Primary | ▼ |
| DE, ES, SE | Household income | Not disclosed | ▼ |
| DE, ES | Diet | 'Other' | ▼ |
| ES | Diet | 'Don't know' | ▼ |
| DE | Ethnicity | South Asian | ▼ |
| DE, ES, SE | Ethnicity | 'Don't know' | ▼ |

*Germany (DE), Spain (ES), Sweden (SE).

**Statistical Test Results.

▲ indicates result is significantly higher than the result in the Total column.

▼ indicates result is significantly lower than the result in the Total column.

Cross-national differences in agreement were small. Any interpretation that such differences reflect national environmental attitudes or policy salience should therefore be treated as speculative. Future research could experimentally test how factors such as quantified evidence, uncertainty ranges, or third-party certification affect belief formation, trust, and engagement outcomes such as willingness to try fungi-based foods [27,39].

### 3.5. Perceptions of fungi-based foods' nutritional value compared to meat

Fungi-based foods are often marketed as protein-rich options that can substitute for meat. To examine public beliefs about this claim, respondents were asked whether fungi-based food can offer a comparable amino-acid composition to meat. This item captures agreement with a broad nutrition claim and should not be interpreted as a comprehensive measure of nutritional knowledge or as evidence of affective, cognitive, or behavioral engagement.

The vast majority of respondents in all three countries endorsed the nutrition statement by responding 'true' (Fig 5; Table 9). Because the item is positively framed and broad, high endorsement may reflect general beliefs or openness to a health-related claim rather than demonstrated nutritional literacy or acceptance. While it should not be interpreted as validation of survey agreement, why respondents may perceive the nutrition-related statement as plausible can be contextualized by pointing to analytical and controlled feeding studies indicating that some fungi-based products can provide high-quality protein with a favorable amino-acid profile, although comparability depends on product formulation and serving size [2,40].

The ceiling effect limits the ability to detect meaningful subgroup differences and increases vulnerability to acquiescence bias. Cross-national differences were statistically significant but very small in magnitude ($\chi^2$=9.0, p_FDR = 0.014; Cramer's V = 0.04).

Further subgroup analyses, however, revealed noteworthy variations (Table 10). Significantly higher proportions of Swedish female respondents; both German and Swedish respondents with tertiary education; both German and Swedish respondents who follow a flexitarian diet; and Spanish respondents with mixed/dual ethnicity endorsed the statement (selected the response option. Significantly lower proportions are observed among Swedish male respondents;

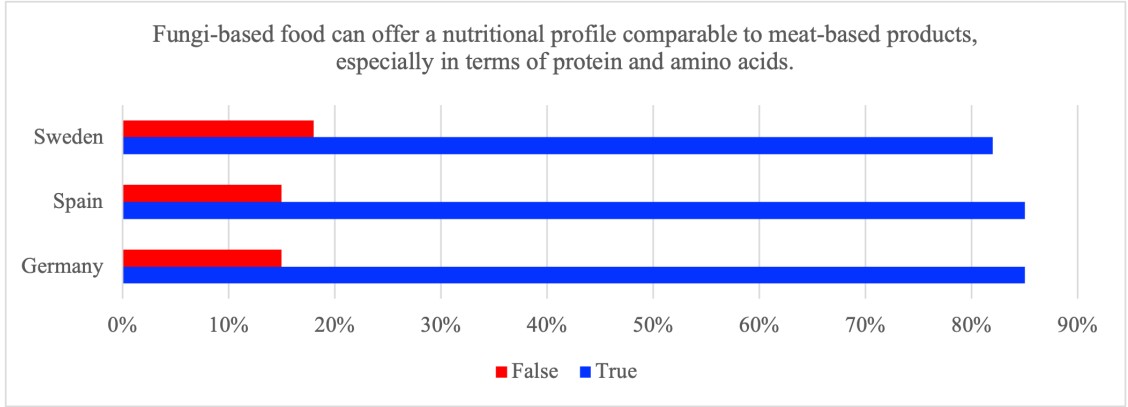

**Fig 5. Proportion of respondents in Germany, Spain and Sweden who answered 'True' or 'False' to the statement 'Fungi-based food can offer a nutritional profile comparable to meat-based products, especially in terms of protein and amino acids.**

**Table 9. Rating profiles and statistically relevant results of the true-or-false statement 'fungi-based food can offer a nutritional profile comparable to meat-based products, especially in terms of protein and amino acids' across Germany (DE), Spain (ES) and Sweden (SE)\*.**

|  | Total | Country | | |
|---|---|---|---|---|
|  |  | DE | ES | SE |
|  |  | A | B | C |
| Base | 6004 | 2002 | 2002 | 2000 |
| True | 84% | 85% | 85% | 82% |
|  |  | C | C |  |
| False | 16% | 15% | 15% | 18% |
|  |  |  |  | A.B |
| Total Sum | 100% | 100% | 100% | 100% |

\* Cell Contents (Column Percentages, Statistical Test Results), Statistics (Column Proportions, (95%): A/B/C, Minimum Base: 30 (\*\*), Small Base: 100 (\*))

▲ indicates result is significantly higher than the result in the Total column.

▼ indicates result is significantly lower than the result in the Total column.

18–24-year-old Swedish respondents; both German and Spanish respondents who wish not to disclose their household income; both German and Swedish respondents who eat meat; Spanish respondents who selected 'do not know' for their diet; and Spanish respondents who selected 'do not know' for their ethnic background.

In terms of interpretation of survey responses, agreement with the nutrition statement indicates endorsement of a broad claim and should not be interpreted as a measure of affective, cognitive, or behavioral engagement or of comprehensive nutritional knowledge. For analytic purposes, 'true' was coded as consistent with the statement's intended emphasis on protein and amino-acid comparability; however, the phrase 'nutritional profile' can reasonably be interpreted more broadly, e.g., as including micronutrients, in which case a 'false' response may reflect a broader evaluative criterion rather than a knowledge deficit. This ambiguity limits the extent to which this item can be treated as a factual knowledge test.

Nutritional equivalence to meat is not one-dimensional. Beyond protein quality, the broader nutritional profile can differ substantially across products, e.g., vitamin $B_{12}$ and heme iron are typically higher in meat, while fiber is higher in fungi-based foods; sodium may vary with processing; and fungi-based biomass is cholesterol-free [41,42]. A nutritionally sophisticated respondent could therefore reasonably disagree with a blanket statement about a 'comparable nutritional

**Table 10. Statistically significant sociodemographic variables regarding the correct answer to the true-or-false statement 'fungi-based food can offer a nutritional profile comparable to meat-based products, especially in terms of protein and amino acids'.**

| Country* | Sociodemographic variable | Subgroup | Direction of significance** |
|---|---|---|---|
| SE | Gender | Female | ▲ |
| DE, SE | Education | Tertiary | ▲ |
| DE, SE | Diet | Flexitarian | ▲ |
| ES | Ethnicity | Mixed/dual ethnicity | ▲ |
| SE | Gender | Male | ▼ |
| SE | Age | 18–24 years | ▼ |
| DE, ES | Household income | Not disclosed | ▼ |
| DE, SE | Diet | Eat meat | ▼ |
| ES | Diet | 'Do not know' | ▼ |
| ES | Ethnicity | 'Do not know' | ▼ |

*Germany (DE), Spain (ES), Sweden (SE)

**Statistical Test Results

▲ indicates result is significantly higher than the result in the Total column.

▼ indicates result is significantly lower than the result in the Total column.

profile' depending on how the term was interpreted. Future work should measure specific nutrient knowledge separately from beliefs or openness to claims and from engagement outcomes, such as willingness to purchase and integrate into dietary routines.

Across domains, subgroup differences were generally small in effect size and should be interpreted cautiously, particularly given the large number of comparisons. Psychological constructs such as gender norms, food technology attitudes, or food technology attitudes, or media exposure were not measured, and therefore observed subgroup patterns cannot be attributed to specific mechanisms. The subgroup analyses were treated as descriptive inputs that may inform future, theory-driven work and the design of targeted communication materials.

## Limitations

This study has several limitations. The five closed-ended items provide a pragmatic, cross-national snapshot of selected misconceptions and claim-related beliefs; they are not a validated psychometric scale of 'science literacy' and primarily capture declarative recognition rather than deeper conceptual understanding or functional application.

Measurement validity is uneven across items. The taxonomy statement 'Fungi and mushrooms are the same thing and are plants' combines two propositions and therefore captures a composite misconception rather than isolating each concept. The mycoprotein statement relied on a simplified definition suitable for survey fielding, but the term is used variably across regulatory and market contexts; responses should therefore be interpreted as familiarity with a common usage rather than knowledge of a single standardized definition. Similarly, the cultivation-timeline statement may be influenced by unfamiliarity with the term 'filamentous fungi' and respondents' tendency to map the question onto mushroom cultivation, meaning that incorrect or uncertain responses may partly reflect semantic ambiguity rather than a precise misconception about fermentation cultivation cycles. Finally, the sustainability and nutrition statements are positively framed and show high agreement, suggesting potential ceiling effects and susceptibility to acquiescence bias; they likely reflect general beliefs or openness to claims rather than tested, in-depth knowledge.

Although quota sampling improves demographic balance, the data come from a proprietary, opt-in online panel and are not probabilistic. Online-only recruitment excludes individuals with limited internet access, and panel membership and

digital engagement may systematically differ from the general population, e.g., in education, income, media use, or interest in new foods, which limits the generalizability of prevalence estimates of knowledge and beliefs. Moreover, repeated participation in panel surveys may introduce panel conditioning, e.g., increased familiarity with survey formats or topics, and post-stratification weights cannot fully remove unobserved selection biases. The sociodemographic profile is provided in Appendix 2 to support evaluation of sample composition.

The analyses involve many subgroup comparisons; while we report effect sizes and apply False Discovery Rate, i.e., Benjamini–Hochberg, adjustment within families of tests, subgroup findings should still be interpreted as exploratory and judged by practical magnitude as well as statistical significance. The cross-sectional design cannot establish causal relationships between knowledge and acceptance; future work should pair validated multi-item knowledge measures with experimental communication interventions to test whether targeted information increases understanding and willingness to try fungi-based foods.

The analysis involves a large number of subgroup comparisons across countries and sociodemographic variables. Although we report effect sizes and apply FDR adjustments for multiple comparisons, residual risk of false positives remains and subgroup findings should be treated as exploratory. Future research should prioritize preregistered hypotheses and parsimonious modeling.

The survey was fielded in three languages. Although professional translation procedures were used, translation- or phrasing-related differences that influence how respondents interpreted key terms, e.g., 'mycoprotein', 'filamentous fungi', and 'nutritional profile,' cannot be ruled out. Cognitive interviews or formal tests of measurement invariance were not conducted, and therefore cross-national comparisons should be interpreted as descriptive patterns rather than definitive evidence of underlying differences in knowledge.

Behavioral outcomes, e.g., engagement with such food through willingness to try, purchase or integrate into dietary routines were not measured nor attitudinal scales validated. Accordingly, the data cannot be used to infer engagement predictions or causal pathways from knowledge and beliefs to behavior.

## Conclusion

Across Germany, Spain, and Sweden, this study provides a cross-national quantification of domain-specific factual knowledge about fungi-based foods. The results show that foundational taxonomy- and process-related misconceptions remain common and socially patterned, while agreement with broad sustainability and nutrition statements is high but likely reflects general beliefs or openness to positive claims rather than tested understanding. Empirically, these findings complement prior attitudinal work on new foods by identifying concrete, communication-relevant knowledge gaps that may condition how information is processed and trusted. Conceptually, the pattern underscores that public engagement involves both informational deficits and credence-based judgments, motivating future research on the mediating roles of exposure, motivation, and trust in shaping engagement-related outcomes.

From a practical perspective, and recognizing that the present data are descriptive, communication strategies may benefit from standardizing terminology, avoiding unnecessary technical language, and addressing the most prevalent misconceptions directly, e.g., clarifying that fungi are neither plants nor animals and explaining basic cultivation timelines. Because the sustainability and nutrition statements are susceptible to ceiling and acquiescence effects, communicators should avoid assuming that agreement reflects stable conviction or actual engagement with such food; instead, messages should transparently specify which outcomes are supported by evidence and under what conditions.

Future work should validate broader multi-item measures of knowledge and beliefs, assess the robustness of subgroup differences under multiple-testing control, and test causal pathways experimentally, e.g., whether targeted information exposure improves understanding, shifts perceived credibility of claims, and influences willingness to try or purchase. Such work would strengthen the evidence base for designing scalable public engagement and communication approaches for fungi-based foods.

## Supporting information

**S1 File. Supplementary materials.** Includes.
(DOCX)

## Author contributions

**Conceptualization:** Coralie Hellwig, Mohammad J. Taherzadeh.

**Data curation:** Coralie Hellwig.

**Formal analysis:** Coralie Hellwig.

**Investigation:** Coralie Hellwig.

**Methodology:** Coralie Hellwig, Mohammad J. Taherzadeh.

**Project administration:** Mohammad J. Taherzadeh.

**Resources:** Mohammad J. Taherzadeh.

**Supervision:** Mohammad J. Taherzadeh.

**Validation:** Coralie Hellwig.

**Visualization:** Coralie Hellwig.

**Writing – original draft:** Coralie Hellwig.

**Writing – review & editing:** Coralie Hellwig, Mohammad J. Taherzadeh.

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
