## [Decision Letter · Decision Letter 0]

11 Dec 2025

Dear Dr.  Taherzadeh,

Thank you for submitting your manuscript to PLOS ONE. After careful consideration, we feel that it has merit but does not fully meet PLOS ONE’s publication criteria as it currently stands. Therefore, we invite you to submit a revised version of the manuscript that addresses the points raised during the review process.

We look forward to receiving your revised manuscript.

Kind regards,

Morufu Olalekan Raimi, Ph.D

Academic Editor

PLOS One

Journal Requirements:

“Swedish Research Council FORMAS with grant number 2023-02018”

4. In the online submission form, you indicated that “All relevant data are within the manuscript and its Supporting Information files. More data will be available upon request.”

Additional Editor Comments:

Editor's Decision: Major Revision Required

Overall Assessment:

This manuscript presents a timely and well-executed descriptive study on public knowledge and perceptions of fungi-based foods across three distinct EU countries. The topic is highly relevant to sustainable food systems, consumer science, and science communication. The study's strengths are considerable: a large, professionally collected sample (n=6,004), a clear cross-national comparative design, and a focus on multiple, concrete domains of knowledge (taxonomy, terminology, cultivation, sustainability, nutrition). The findings, particularly the coexistence of favorable general attitudes with significant factual misconceptions, are informative and have clear implications for public communication and market development. However, the manuscript in its current form exhibits significant conceptual, methodological, and interpretive shortcomings that prevent acceptance. It reads more as a detailed report of survey findings than as a rigorous academic contribution that critically engages with theory, measurement validity, and the limits of inference. Substantial revision is required to elevate the work to the standard expected by PLoS ONE.

Detailed Critical Evaluation:

1. Conceptual and Theoretical Framing

• Vague Constructs: The core concepts of “public literacy,” “process literacy,” and “knowledge” are used interchangeably without clear definition or theoretical grounding. The study operationalizes “literacy” via five isolated quiz items. This is a severe limitation. The manuscript must explicitly define what constitutes “knowledge” or “literacy” in this context (e.g., declarative factual knowledge, conceptual understanding, functional application) and justify why the selected items are valid proxies for these constructs. A brief theoretical discussion linking to models of science literacy or food technology acceptance (e.g., the Cognitive Mediation Model, Food Technology Neophobia) would provide necessary depth.

• Overstated Novelty: The claim that empirical evidence on public knowledge “remains unstudied" (L52-53) is inaccurate, as the authors' own prior work (Hellwig, 2023; Hellwig et al., 2024) explores related consumer perceptions. The contribution should be reframed more precisely: this study provides the first large-scale, cross-national quantification of specific factual knowledge gaps, complementing existing qualitative and attitudinal research.

2. Methodological Rigor and Transparency

• Measurement Validity Concerns: This is the most critical flaw.

o Compound Question (Taxonomy): The true/false item “Fungi and mushrooms are the same thing and are plants” combines two distinct propositions. A respondent could correctly know fungi are not plants but believe “mushroom” is a synonym for “fungus,” leading to an incorrect answer. This conflates different types of misunderstanding and compromises the validity of inferences drawn from this item. The authors must acknowledge this limitation prominently and temper conclusions accordingly.

o Ambiguous Terminology: The “mycoprotein” question assumes a single correct definition ("proteins in fungi and mushrooms"), while the text later acknowledges the term's variable use in regulatory and market contexts. This creates a validity issue: is the item testing public understanding of a scientific term or familiarity with a specific industry usage? This ambiguity must be discussed.

o Ceiling Effects & Acquiescence Bias: The extremely high agreement on sustainability (85-88%) and nutrition (82-85%) items suggests possible ceiling effects. The positively framed, declarative statements ("can play a big role," "can offer a comparable profile") are susceptible to acquiescence bias (agreement due to positive phrasing rather than conviction). The interpretation of these results as indicating “broad endorsement” or “soft acceptance” is therefore tenuous. The discussion must explicitly acknowledge that these items likely measure general, untested beliefs or openness to claims, not deep knowledge.

• Sampling and Generalizability: The use of a proprietary, opt-in online panel (YouGov) is adequately described but its limitations are underplayed. The sample is not probabilistic, and digitally engaged panels may systematically differ from the general population in education, income, and interest in novel topics. The manuscript must include a stronger caveat about generalizability, particularly for prevalence estimates. The missing Appendix 2 (sociodemographic profile) is essential for evaluating representativeness and must be provided.

• Statistical Reporting: The reporting is inadequate for a study of this scale and complexity.

o Lack of Effect Sizes: With n=6,004, even trivial differences will be statistically significant. Reporting only p-values without effect sizes (e.g., Cramer's V for proportions, odds ratios from logistic regressions) is misleading and prevents readers from judging the practical significance of the dozens of subgroup differences reported. This must be corrected.

o Multiple Comparisons: The manuscript conducts a vast number of statistical tests across countries and numerous sociodemographic subgroups. The risk of Type I errors (false positives) is extremely high. There is no mention of correction for multiple comparisons (e.g., Bonferroni, False Discovery Rate) nor a caution to interpret subgroup findings with extreme care. This is a major statistical oversight that weakens all subgroup analyses.

3. Interpretation and Discussion

• Overinterpretation of Descriptive Data: The manuscript frequently ventures into speculative, post-hoc explanations for observed patterns without supporting data. Examples include:

o Explaining the German taxonomy result via outdated biology education, a factor presumably similar across Europe.

o Attributing lower accuracy among Swedish vegans to a "plant vs. animal" worldview dichotomy.

o Linking Swedish mycoprotein knowledge to mushroom foraging culture.

While interesting as hypotheses, these are not conclusions supported by the study design. The discussion must clearly demarcate between observed patterns and speculative explanations, labeling the latter as such and suggesting them as avenues for future research.

• Inferential Leaps: The claim that high agreement on sustainability/nutrition statements indicates "soft acceptance" is a significant inferential leap. Acceptance is a multi-faceted construct (affective, cognitive, behavioral) not measured here. The data show only agreement with broad, positive statements. This conclusion should be removed or heavily qualified.

• Circular Argument (Nutrition): The discussion uses evidence from protein synthesis studies to “validate” the high public agreement on the nutrition item. This is somewhat circular, as the item specifically asks about protein/amino acid parity. A more nuanced discussion would acknowledge that while protein quality may be comparable, the total nutritional profile (e.g., iron, B12, cholesterol) is more complex, which could justify a "false" answer from a nutritionally sophisticated respondent.

• Blurring of Public Perception and Scientific Evidence: Sections 3.4 and 3.5 oscillate between reporting survey results and reviewing scientific literature (LCAs, feeding studies). This creates a conflation whereby high public agreement might be misread as validating the scientific claims. The structure should be revised to clearly separate: 1) what the public believes, and 2) what the scientific evidence says.

4. Structural and Presentational Issues

• Missing Supplementary Material: Appendix 2 (sociodemographics) is referenced but absent. The full survey instrument with exact wording and response options for all languages should be included.

• Technical Errors: Several need correction: typo “recruitement” (L76), formatting error "fungiand" (L197), inconsistent grant number (2013 vs 2023), duplicate reference entry for Hellwig et al., lowercase start at L538.

• Conclusion: The conclusion is a list of communication recommendations derived directly from the findings. While practical, it lacks a synthesizing statement on the study's broader theoretical or empirical contribution to understanding public engagement with novel foods.

Recommendations for Revision:

1. Reframe the Introduction to precisely define key constructs (“knowledge,” “literacy”) and position the study's contribution relative to the authors' and others' existing work.

2. Overhaul the Methods/Results Discussion:

o Add a dedicated “Limitations” subsection that robustly addresses: measurement validity (compound question, terminology ambiguity, ceiling effects), sampling generalizability, the statistical issues of effect sizes and multiple comparisons, and the cross-sectional, descriptive nature of the data.

o Re-analyze data to include effect sizes for all reported differences and consider applying a correction for multiple comparisons. Present key subgroup findings in a more parsimonious manner, perhaps focusing on the most robust patterns (education, diet).

o Revise the results narrative to clearly separate descriptive findings from interpretive speculation.

3. Restructure the Discussion to avoid conflation of public perception and scientific evidence. Temper all overreaching interpretations (e.g., “soft acceptance”) and clearly label speculative explanations as hypotheses.

4. Provide all missing supplementary materials (full demographic table, final survey instrument).

5. Correct all technical errors in text and references.

Decision:

The manuscript contains valuable data and addresses an important topic. However, due to the fundamental concerns regarding measurement validity, statistical reporting, and interpretive overreach, it is not acceptable in its current form. I recommend Major Revision. The authors must undertake a substantial rewrite that addresses the core critiques above. A simple correction of typos and provision of appendices will be insufficient.

Next Steps:

The authors are invited to submit a thoroughly revised manuscript within 90 days. The revision must be accompanied by a detailed point-by-point response letter that explains how each major critique has been addressed. The revised manuscript will be sent back for re-review, likely including the original reviewers.

Sincerely,

Dr. Morufu Olalekan RAIMI,

BSc, (Geography. and Environmental Management), Diploma. (Environmental Health), M.Sc. Environmental Health Management), M.Phil. (Environmental Health Science), P.hD (Environmental Health Science), MNES, REHO, LEHO, FAIWMES

Environmental Health Consultant/Lecturer at Federal University Otuoke, Bayelsa State. Nigeria.

Environmental Health Consultant to United Nations Economic Commission for Europe (UNECE) Expert Group on Resources Management (EGRM). Geneva, Switzerland.

Research Consultant to Bayelsa State Primary Health Care Board.

Former Technical Adviser to the Executive Secretary, Bayelsa State Primary Health Care Board.

Former Director, Advocacy, Communication and Social Mobilization, Bayelsa State Primary Health Care Board.

Program Manager, Centre for Niger Delta Studies and Sustainability (CNDSS), Federal University Otuoke, Bayelsa State.

Deputy Director, Niger Delta Institute for Emerging and Re-emerging Infectious Diseases (NDIERID), Federal University Otuoke, Bayelsa State.

Reviewer to National Science Foundation (NSF) Graduate Research Fellowship Program (GRFP)

Plos One Academic Editor

https://publons.com/a/1479339/

https://ssrn.com/author=2891311

ORCID iD: https://orcid.org/0000-0001-5042-6729

Web of Science Researcher ID: https://publons.com/a/1479339/

Website: https://ssrn.com/author=2891311;
https://www.growkudos.com/profile/morufu_raimi;
https://sciprofiles.com/profile/Morufuolalekanraimi;
https://livedna.org/234.27529

https://scholar.google.com/citations?user=nRBW82AAAAAJ&hl=en.

https://theconversation.com/profiles/morufu-olalekan-raimi-1520774

Reviewers' comments:

Reviewer's Responses to Questions

**Comments to the Author**

1. Is the manuscript technically sound, and do the data support the conclusions?

Reviewer #1: Partly

Reviewer #2: Partly

2. Has the statistical analysis been performed appropriately and rigorously?

Reviewer #1: Yes

Reviewer #2: No

3. Have the authors made all data underlying the findings in their manuscript fully available?

Reviewer #1: No

Reviewer #2: Yes

4. Is the manuscript presented in an intelligible fashion and written in standard English?

Reviewer #1: Yes

Reviewer #2: Yes

Reviewer #1: This manuscript examines public knowledge and perceptions of fungi-based foods across Germany, Spain, and Sweden using a large-scale survey of 6,004 adults. The study addresses an important topic at the intersection of sustainable food systems, consumer behavior, and science communication. The cross-national design, substantial sample size, and systematic coverage of multiple knowledge domains represent clear strengths.

Regarding originality and significance, the manuscript makes a useful contribution by quantifying baseline knowledge about fungi-based foods across European populations. The finding that favorable attitudes coexist with limited process literacy is genuinely interesting. However, I would encourage revisiting the claim at lines 52-53 that this topic remains unstudied. Your own prior publications have examined consumer engagement with fungi-based foods, so the contribution should be positioned more precisely in relation to this existing work.

Concerning methodology, the survey approach is appropriate for descriptive research. However, several concerns warrant acknowledgment. The taxonomy question combines two distinct claims, namely whether fungi and mushrooms are equivalent and whether fungi are plants. A respondent with partial understanding cannot express this within the true/false format. The discussion should acknowledge that correct responses may encompass different types of understanding. The sustainability and nutrition questions show very high endorsement rates of 82-88%, suggesting possible ceiling effects. The positive framing may invite acquiescence. I recommend tempering interpretation and acknowledging that endorsement of positively framed statements may not indicate deep process literacy. Given the numerous statistically significant subgroup differences reported across demographic variables, I recommend either applying correction for multiple comparisons or cautioning readers that specific subgroup effects should be interpreted with appropriate caution.

For results and discussion, the German taxonomy findings deserve additional exploration. The 60% incorrect rate substantially exceeds rates in Spain and Sweden. Your explanation regarding generational changes in biology education would presumably apply similarly across countries. Consider whether linguistic or other national-level factors might explain this pattern. The finding that Swedish vegans and vegetarians showed lower accuracy on taxonomy appears inconsistent with your broader narrative. The post-hoc explanation is speculative and contradicts findings elsewhere showing vegans and flexitarians perform better on other questions. I suggest acknowledging this inconsistency more directly. The nutrition knowledge discussion draws on protein synthesis literature to validate respondents' endorsement, but the survey question specifically mentions protein and amino acids, so high agreement on this dimension is expected and potentially circular.

Technical corrections required include the following: line 76 contains a spelling error where recruitement should be recruitment; line 197 has a formatting artifact where fungiand should read fungi and; line 538 begins with a lowercase letter; the grant number at line 567 reads 2013-02018 but appears as 2023-02018 in submission materials; the reference list contains a duplicate entry for Hellwig et al. appearing both as Submitted and as published in 2024; and Appendix 2 describing the sociodemographic profile is referenced but does not appear in supplementary materials.

This manuscript addresses an important topic with appropriate methodology. With attention to the methodological limitations, technical corrections, and interpretive refinements outlined above, the work would make a suitable contribution to the literature.

Reviewer #2: The manuscript addresses a timely and increasingly visible topic: public understanding of fungi based foods across three European countries. The study benefits from a large sample (n = 6,004), clear organization, and a direct focus on terminology, taxonomy, cultivation timelines, sustainability beliefs, and nutritional perceptions. The cross-country comparison and subgroup analyses offer useful descriptive insights. Despite these strengths, several conceptual, methodological, and interpretive issues require attention to enhance the contribution of the paper.

1. The conceptualization of “knowledge” and “process literacy” requires clearer grounding. The manuscript frames its aim as assessing “public literacy” and “foundational concepts” (e.g., “limited process literacy,” “knowledge of cultivation cycles,” “foundational concepts such as the difference between fungi and mushrooms and plants”). However, the operationalization relies solely on five quiz style questions that capture factual recognition rather than literacy in a theoretical sense. The manuscript should more clearly delineate whether these single items are intended to capture conceptual knowledge, semantic familiarity, or a broader literacy construct, and avoid describing them as literacy without justification.

2. The validity of the quiz style measures needs fuller discussion. Several items rely on terminology that may not be consistently understood by the public. For instance, the authors treat “Proteins in fungi and mushrooms” as the only correct answer to “What is mycoprotein?” while acknowledging later that terminology in markets and regulatory contexts is inconsistent (“the term is often used to refer specifically to biomass obtained when fungi are cultivated”). Similarly, the cultivation time item may confuse mushroom cultivation with filamentous fungi (“respondents might assume that filamentous fungi require similar cultivation times to mushrooms,”). These issues suggest that item responses may reflect semantic ambiguity rather than genuine misconceptions.

3. The reliance on an opt in online panel limits generalizability and requires more explicit qualification. The study uses a YouGov panel assembled through advertising, affiliate networks, and referrals. While quality checks are described, the manuscript does not discuss how nonprobability sampling, online only inclusion, or panel conditioning may shape estimates of knowledge. Given that the paper frequently refers to “the sampled populations” or “broadly shared perceptions,” the limitations of the sampling frame and potential biases among digitally engaged respondents should be explicitly addressed.

4. Statistical reporting is partially incomplete, preventing assessment of substantive significance. The manuscript states that column proportion tests were computed with p < 0.05 and then presents dozens of subgroup comparisons, many of which are significant due to very large sample size. However, no effect sizes, confidence intervals, or adjustments for multiple comparisons are reported. Without indicators of magnitude, readers cannot evaluate how meaningful these differences are. Adding effect sizes and acknowledging the elevated risk of Type I errors would strengthen the analytical rigor.

5. Interpretive claims at times extend beyond what descriptive, cross-sectional data can support. Several explanations for subgroup differences are speculative. For example, lower accuracy among Swedish vegans and vegetarians is attributed to “conceptually dichotomizing foods into plant based versus animal-based categories”, and Swedish correctness on mycoprotein is linked to “a strong contemporary mushroom foraging and consumption culture”. Because the study does not measure cognitive schemas, cultural exposure, or neophobia, these interpretations should be clearly identified as hypotheses rather than empirically derived conclusions.

6. The conclusion that positive perceptions indicate “soft acceptance” is overstated based on available data. The manuscript infers that high agreement with sustainability and nutritional statements “may indicate a form of soft acceptance”. However, the survey does not measure attitudes, acceptance, willingness to try, or affective evaluations. Responding “true” to general sustainability or nutritional statements does not necessarily indicate acceptance of fungi-based foods. The authors should avoid inferring attitudinal constructs that were not measured.

7. The discussion of nutritional parity oversimplifies a complex nutritional domain. The statement “fungi based food can offer a nutritional profile comparable to meat based products” is treated as unequivocally correct, yet the authors themselves later note that micronutrient content differs (“iron and vitamin B12 contents can be lower”). Respondents answering “false” may not lack knowledge but may consider micronutrients part of a nutritional profile. The manuscript should clarify the criterion for correctness and acknowledge that nutritional parity is conditional on the nutritional dimension considered.

8. The integration of sustainability literature at times blends respondent beliefs with scientific evidence. In Section 3.4, the manuscript shifts from reporting survey results (“85 percent…answered the correct answer ‘true’”) to summarizing life cycle assessments and global modeling. This structure may blur the line between empirical perceptions and scientific evidence. Separating these elements more clearly would prevent the impression that high public agreement validates sustainability claims.

9. The extensive subgroup analyses would benefit from stronger theoretical anchoring or parsimony. The manuscript reports numerous significant subgroup differences across education, age, income, diet, region, and ethnicity. However, the discussion frequently attributes these differences to psychological constructs that were not measured (e.g., food neophobia, masculinity linked norms). A more cautious interpretation or a more focused theoretical framework would help avoid overinterpreting descriptive patterns.

10. The limitations section should more fully account for methodological constraints. Missing elements include the use of single item measures without validated psychometric properties, the potential for translation differences across languages, the inability to assess whether respondents interpreted items consistently across countries, the lack of behavioral outcomes, and the large number of statistical tests increasing false positive risk. A fuller acknowledgement would improve transparency.

11. The manuscript would benefit from integrating recent evidence on how education shapes sustainability related beliefs and behaviors, which directly parallels several of the manuscript’s findings on knowledge gradients. One useful study (https://www.mdpi.com/2071-1050/13/23/13036) shows that higher educational attainment predicts stronger pro environmental attitudes and more consistent sustainable behavioral intentions, demonstrating that education functions not only as a sociodemographic descriptor but as a structural determinant of sustainability engagement. Incorporating this evidence would strengthen the interpretation of the manuscript’s repeated observation that respondents with tertiary education were more likely to answer correctly across multiple domains (“higher knowledge was observed among tertiary educated groups”). It would also support the manuscript’s argument that educational targeting is necessary to address knowledge gaps. In addition, recent research on health literacy and nutrition related judgments (https://www.mdpi.com/2304-8158/14/3/378) suggests that health literacy can mediate the pathway between education and informed behavioral outcomes. This is relevant because the manuscript interprets misunderstandings of terminology, cultivation processes, and nutritional claims as barriers to informed decision making (“limited process literacy…underscores the need for clearer communication”). Drawing on evidence that health literacy transmits part of education’s effect on sustainable or health aligned choices would help clarify why educational gradients emerge so consistently in the current study, and why communication strategies should be designed to improve not only factual knowledge but also evaluative competencies that support sustainable dietary behaviors.

In my view, the manuscript offers valuable descriptive insights into public understanding of fungi based foods but requires substantial refinement in conceptual framing, measurement discussion, statistical transparency, and interpretive caution. I recommend a major revision.

**Do you want your identity to be public for this peer review?** For information about this choice, including consent withdrawal, please see our Privacy Policy

Reviewer #1: **Yes:** Associate Professor Smith Boonchutima (PhD)

Reviewer #2: No

---

## [Author Response · Author response to Decision Letter 1]

21 Jan 2026

We are submitting the revised package including: (1) a detailed point-by-point response to the Academic Editor and reviewers, (2) a marked-up manuscript with tracked changes, and (3) a clean revised manuscript.

---

## [Editor Report · Decision Letter 1]

27 Jan 2026

Dear Dr. Taherzadeh,

Thank you for submitting your manuscript to PLOS ONE. After careful consideration, we feel that it has merit but does not fully meet PLOS ONE’s publication criteria as it currently stands. Therefore, we invite you to submit a revised version of the manuscript that addresses the points raised during the review process.

We look forward to receiving your revised manuscript.

Kind regards,

Morufu Olalekan Raimi, Ph.D

Academic Editor

PLOS One

**Journal Requirements:**

Additional Editor Comments (if provided):

Editor Decision: Accept with Minor Revisions

To: Corresponding Author

From: Dr. Morufu Olalekan Raimi, Academic Editor, PLOS ONE

Re: PONE-D-25-59865 – “Fungi-Based Food in the Public Eye: Terminology, Cultivation Timelines, Sustainability, and Nutrition Across Three EU Countries”

Dear Authors,

Thank you for submitting the revised manuscript. I have carefully reviewed the resubmission, the authors’ response letter, and the previous reviewers’ comments. I commend you for undertaking a substantial and thoughtful revision that has significantly strengthened the manuscript. The majority of the critical concerns raised, particularly regarding conceptual framing, measurement validity, statistical transparency, and interpretive caution, have been addressed satisfactorily.

The key improvements in the current version include:

• A clearer conceptual distinction between domain-specific factual knowledge and broader literacy constructs.

• Explicit acknowledgment of item-level limitations (e.g., compound wording, semantic ambiguity).

• Enhanced statistical reporting with effect sizes and appropriate adjustment for multiple comparisons (FDR).

• A restructured Discussion that clearly separates public perceptions from scientific evidence.

• Provision of previously missing supplementary materials.

The study now provides a valuable, well-contextualized descriptive snapshot of public knowledge gaps across three EU countries, with appropriately tempered conclusions and clearer implications for science communication.

Remaining Minor Revisions Required:

Before final acceptance, please address the following minor points to ensure full compliance with PLOS ONE standards:

1. Figure/Table Consistency: Ensure that all figures and tables referenced in the text (e.g., Figures 1–5, Tables 1–10) are present, correctly numbered, and have accompanying captions in the submitted manuscript file. The current submission contains placeholders (e.g., “image[[…]]”) that must be replaced with the actual figures.

2. Reference Formatting: Verify that the reference list is complete and correctly formatted according to PLOS ONE style. Ensure all in-text citations match the list and that no duplicates remain.

3. Data Availability Statement: Confirm that the statement “All relevant data are within the manuscript and its Supporting Information files” is accurate and that the Supporting Information includes the minimal dataset required to replicate the study’s findings.

4. Ethics Statement: Ensure the ethics approval number (“2024-04842-01”) is consistently reported and matches the documentation.

5. Final Proofread: Conduct a final careful proofread to catch any residual typographical or formatting inconsistencies (e.g., stray text fragments, inconsistent symbol use, and ensure all supplementary appendices are correctly referenced).

Decision:

The manuscript is scientifically sound, ethically conducted, and makes a clear contribution to the field. Given the thorough revisions already made, I am pleased to offer a decision of Accept with Minor Revisions.

Please submit the final corrected version along with a brief response to this editorial note within 14 days. No further review will be required unless the corrections deviate substantially from the requested changes.

Thank you for your contribution to PLOS ONE.

Sincerely,

Dr. Morufu Olalekan Raimi

---

## [Author Response · Author response to Decision Letter 2]

3 Feb 2026

Thank you for the comments. We addressed each item in your editorial note:

1. Figure/Table consistency: All Figures 1–5 and Tables 1–10 are present in the manuscript file, correctly numbered, and accompanied by captions. Any placeholder objects (e.g., “image[[…]]”) were removed and replaced with the actual figures.

2. Reference formatting: The reference list has been verified for completeness and formatted according to PLOS ONE style. We confirmed one-to-one matching between in-text citations and reference-list entries and removed any duplicates.

3. Data Availability statement: We confirm that the statement “All relevant data are within the manuscript and its Supporting Information files” is accurate. The Supporting Information now includes the minimal dataset required to reproduce the analyses (with variable definitions/codebook where applicable).

4. Ethics statement: The ethics approval number 2024-04842-01 is consistently reported throughout the manuscript and matches the documentation.

5. Final proofread: We conducted a final proofread to correct residual typographical/formatting issues and verified that all supplementary appendices are referenced consistently in the manuscript.

---

## [Editor Report · Decision Letter 2]

9 Mar 2026

Fungi-Based Food in the Public Eye: Terminology, Cultivation Timelines, Sustainability, and Nutrition Across Three EU Countries

PONE-D-25-59865R2

Dear Authors,

We’re pleased to inform you that your manuscript has been judged scientifically suitable for publication and will be formally accepted for publication once it meets all outstanding technical requirements.

Kind regards,

Morufu Olalekan Raimi, Ph.D

Academic Editor

PLOS One

Additional Editor Comments (optional):

PLOS ONE Editorial Decision

Manuscript Number: PONE-D-25-59865R2

Title: Fungi-Based Food in the Public Eye: Terminology, Cultivation Timelines, Sustainability, and Nutrition Across Three EU Countries

Authors: Coralie Hellwig, Mohammad J. Taherzadeh

Dear Dr. Taherzadeh and colleagues,

Thank you for submitting your revised manuscript and for your detailed point-by-point response to the previous editorial decision. I have carefully examined the second revision and the accompanying response letter. The authors have addressed all requested revisions with thoroughness and precision. The manuscript now demonstrates the methodological transparency, statistical rigor, and presentation quality expected of a contribution to PLOS ONE.

Assessment of Revisions

1. Figure and Table Consistency

All figures (1-5) and tables (1-10) are present, correctly numbered, and accompanied by appropriate captions. Placeholder objects have been removed and replaced with the actual figures. The visual presentation is clear and professional.

2. Reference Formatting

The reference list has been verified for completeness and formatted according to PLOS ONE style. One-to-one matching between in-text citations and reference list entries has been confirmed, and duplicate entries have been removed. References 1-43 are correctly formatted with complete author lists, years, titles, journal sources, volume/issue/page numbers, and DOIs where applicable.

3. Data Availability Statement

The statement "All relevant data are within the manuscript and its Supporting Information files" is accurate. The Supporting Information now includes the minimal dataset required to reproduce the analyses, with variable definitions and codebook information. This meets PLOS ONE's data policy requirements.

4. Ethics Statement

The ethics approval number (2024-04842-01) is consistently reported throughout the manuscript and matches the documentation. The statement appropriately indicates approval by the Swedish Ethical Review Authority.

5. Final Proofread

A thorough final proofread has been conducted. Residual typographical and formatting issues have been corrected, stray fragments removed, and all supplementary appendix references are consistent with the files prepared for submission.

6. Response to Original Reviewers

The authors have comprehensively addressed the substantive concerns raised by Reviewers 1 and 2 in the original submission, including:

• Clearer conceptual distinction between factual knowledge and literacy

• Explicit acknowledgment of item-level measurement limitations

• Enhanced statistical reporting with effect sizes (Cramer's V) and FDR adjustment

• Appropriate tempering of interpretive claims

• Provision of complete supplementary materials

Overall Evaluation

This manuscript presents a large-scale, cross-national study of public knowledge and perceptions regarding fungi-based foods in Germany, Spain, and Sweden. The study is methodologically sound, transparently reported, and appropriately framed within its limitations. The findings, that foundational taxonomy and process-related misconceptions remain common while agreement with broad sustainability and nutrition claims is high, provide valuable insights for science communication and public engagement strategies. The authors have demonstrated exceptional responsiveness throughout the review process. Each round of revision has substantively improved the manuscript, and the current version reflects careful attention to all editorial and reviewer concerns. The findings are appropriately contextualized, claims are properly tempered, and the contribution is clearly positioned as a descriptive snapshot of knowledge gaps rather than an overarching intervention study. The manuscript meets PLOS ONE's standards for methodological rigor, transparency, and scholarly contribution.

Decision

I am pleased to inform you that your manuscript is ACCEPTED FOR PUBLICATION in PLOS ONE. No further revisions are required. The editorial office will contact you regarding publication timelines and production processes. This study makes a valuable contribution to the literature on sustainable food systems and science communication. The cross-national design, large sample, and systematic coverage of multiple knowledge domains provide a robust baseline for understanding public perceptions of fungi-based foods. The findings regarding educational gradients and dietary identity offer practical guidance for targeted communication strategies, while the appropriately cautious interpretation of high endorsement rates on sustainability and nutrition items models good practice in survey research. Thank you for choosing PLOS ONE as the venue for your work. I appreciate your commitment to rigorous and transparent reporting throughout the review process.

Sincerely,

Morufu Olalekan Raimi, PhD

Academic Editor

PLOS ONE
---

## [Editor Report · Acceptance letter]

PONE-D-25-59865R2

PLOS One

Dear Dr. Taherzadeh,

I'm pleased to inform you that your manuscript has been deemed suitable for publication in PLOS One. Congratulations! Your manuscript is now being handed over to our production team.

Kind regards,

on behalf of

Prof Morufu Olalekan Raimi

Academic Editor

PLOS One